# TGF-β Isoforms Affect the Planar and Subepithelial Fibrogenesis of Human Conjunctival Fibroblasts in Different Manners

**DOI:** 10.3390/biomedicines11072005

**Published:** 2023-07-15

**Authors:** Megumi Watanabe, Yuri Tsugeno, Tatsuya Sato, Araya Umetsu, Nami Nishikiori, Masato Furuhashi, Hiroshi Ohguro

**Affiliations:** 1Departments of Ophthalmology, School of Medicine, Sapporo Medical University, Sapporo 060-8556, Japan; watanabe@sapmed.ac.jp (M.W.); yuri.tsugeno@gmail.com (Y.T.); araya.umetsu@sapmed.ac.jp (A.U.); megumi.h@sapmed.ac.jp (N.N.); 2Departments of Cardiovascular, Renal and Metabolic Medicine, Sapporo Medical University, Sapporo 060-8556, Japan; satatsu.bear@gmail.com (T.S.); furuhasi@sapmed.ac.jp (M.F.); 3Departments of Cellular Physiology and Signal Transduction, Sapporo Medical University, Sapporo 060-8556, Japan

**Keywords:** TGF-β isoform, human conjunctival fibroblast, 3D culture, conjunctival fibrosis

## Abstract

Three highly homologous isoforms of TGF-β, TGF-β-1~3, are involved in the regulation of various pathophysiological conditions such as wound healing processes in different manners, despite the fact that they bind to the same receptors during their activation. The purpose of the current investigation was to elucidate the contributions of TGF-β-1 ~3 to the pathology associated with conjunctiva. For this purpose, the biological effects of these TGF-β isoforms on the structural and functional properties of two-dimensional (2D) and three-dimensional (3D) cultured human conjunctival fibroblasts (HconF) were subjected to the following analyses: 1) transendothelial electrical resistance (TEER), a Seahorse cellular metabolic measurement (2D), size and stiffness measurements of the 3D HTM spheroids, and the qPCR gene expression analyses of extracellular matrix (ECM) components (2D and 3D). The TGF-β isoforms caused different effects on the proliferation of the HconF cell monolayer evaluated by TEER measurements. The differences included a significant increase in the presence of 5 ng/mL TGF-β-1 and -2 and a substantial decrease in the presence of 5 ng/mL TGF-β-3, although there were no significant differences in the response to the TGF-β isoforms for cellular metabolism among the three groups. Similar to planar proliferation, the TGF-β isoforms also induced diverse effects toward the mechanical aspects of 3D HconF spheroids, where TGF-β-1 increased stiffness, TGF-β-2 caused no significant effects, and TGF-β-3 caused the downsizing of the spheroids and stiffness enhancement. The mRNA expression of the ECMs were also modulated in diverse manners by the TGF-β isoforms as well as the culture conditions for the 2D vs. 3D isoforms. Many of these TGF-β-3 inducible effects were markedly different from those caused by TGF-β1 and TGF-β-2. The findings presented herein suggest that the three TGF-β isoforms induce diverse and distinctly different effects on cellular properties and the expressions of ECM molecules in HconF and that these changes are independent of cellular metabolism, thereby inducing different effects on the epithelial and subepithelial proliferation of human conjunctiva.

## 1. Introduction

It is well recognized that fibroblasts play an important role in maintaining the structural integrity of most tissues and organs by producing the extracellular matrix (ECM) molecules that make up connective tissue. Therefore, it has been shown that fibroblasts may differentiate their functional properties to suitably fit with the origin of the organ and tissue, the location within the body, as well as the spatial distribution [1]. In fact, diversity in gene expression has been shown between derm-related and non-derm-related fibroblasts [2,3]. Furthermore, fibroblasts belonging to different anatomical locations also have different origins during developmental processes, such as the neural crest, lateral plate mesoderm, and others [4,5,6]. Alternatively, fibroblasts originating from different anatomical locations are also associated with potentially different disease pathogenesis such as keloid scar formation [7].

Within the ocular surface tissues, the conjunctiva physiologically contributes to maintaining a healthy condition as well as biological barrier functions. However, within some pathological conditions, including several ocular surface diseases, and surgical interventions, unfavorable fibrogenetic changes are often associated with the superficial and subepithelial locations of the conjunctiva, in which conjunctival fibroblasts (conF) are identified as the responsible cells within these changes [8]. Therefore, for instance, in some cases of ocular surgery, to obtain preferable surgical outcomes for certain types of ocular-surface-related diseases, including glaucoma, it becomes necessary to appropriately monitor and control such subconjunctival fibrosis [9,10,11,12,13,14,15]. In general, concerning the responsible mechanisms related to both physiological and pathological fibrogenic changes, that is, normal wound healing and scar formation, it has been suggested that the effects of several cytokines and growth factors on the fibroblasts are pivotally involved [16,17]. Among these factors, it is well recognized that transforming growth factor-beta (TGF-β) family proteins function to control most of all of the physiological and pathological wound-healing-related processes [17]. Within these processes, TGF-β has been shown to stimulate the so-called “epithelial mesenchymal transition (EMT)”, that is, the trans-differentiation of fibroblasts into myofibroblasts [16,18,19]. It has also been shown that myofibroblasts can be removed from the wound site by apoptosis during the normal physiological wound-healing process. However, in contrast, if such a normal wound-repairing process occurs improperly, pathological fibrosis could be evoked by the myofibroblasts that are produced, leading to unfavorable scar formation [18,20]. Thus, preventing such conversion of fibroblasts into myofibroblasts may be pivotal and a possible strategy to maintain the healthy epithelial and subepithelial conditions of the human conjunctiva [14,21,22]. 

TGF-β is well recognized as a prototype of the TGF-β family that includes three TGF-β isoforms (TGF-β-1, -2, and -3), activin, nodal, bone morphogenetic proteins (BMPs), growth and differentiation factors (GDFs), and other factors [23]. It has been shown that although TGF-β-1, -2, and -3 share approximately 70–80% homology in their amino acid compositions [24] and could exert biological effects through the same receptors as well as the Smad and non-Smad signaling pathways, these three isoforms are differently involved in the regulation of the wound healing processes [25]. Therefore, these TGF-β isoforms are promising therapeutic target factors for the pathogenic states of the wound healing processes. For instance, the inhibition of TGF-β/Smad3 signaling caused a significant reduction in wound scarring formation, thus providing an expectation of satisfactory wound healing [26,27]. However, since TGF-β-1 or -β-2 could stimulate ECM deposition during the early phases of wound healing processes, the final consequence of wound scarring was not different between untreated wounds or wounds treated with TGF-β-1 or -2 [25]. Alternatively, TGF-β-3 causes a significant reduction in cutaneous scar formation [25], and thus, a recombinant TGF-β-3 ligand was used for its prevention in a phase I/II clinical trial [28]. Quite interestingly, it was revealed that these three TGF-β isoforms were identified within the aqueous humor (AH) [29], and all of them may contribute to the pathogenesis of glaucoma, but in different manners [30,31,32]. In addition, since AH perfuses into the subconjunctival space in the case of glaucoma filtering surgeries, understanding the effects of TGF-β isoforms against conjunctiva would be of outstanding interest. For example, among the TGF-β isoforms, previous studies have revealed that (1) TGF-β-1 induces collagen production in the conjunctival fibroblast which is mediated by NADPH oxidase 4 (Nox4)-related signaling [33], (2) the amniotic membrane matrix uniquely suppresses the production of TGF-β-2 and -3 [34], (3) TGF-β signaling inhibits goblet cell differentiation via the SAM-pointed domain that contains the ETS transcription factor (SPDEF) in the conjunctival epithelium [35], and (4) TGF-β signaling plays a significant profibrotic role in the pterygium via stimulating EMT [36,37]. Thus, these collective observations suggest that further investigations concerning the functional diversity among these TGF-β isoforms may facilitate not only a better understanding of pathophysiological conjunctival fibrosis but would also provide some clues for developing new therapeutic strategies for the treatment of pathogenic conjunctival fibrosis. 

In the current investigation, to study the influences that the TGF-β isoforms induce on the human conjunctiva, we used two-dimensional (2D) and three-dimensional (3D) cultures of human conjunctival fibroblast (HconF) cells, which have recently been reported to be suitable in vitro conjunctival models for epithelial and subepithelial fibrogenetic changes, respectively [38]. 

## 2. Materials and Methods

The current study was conducted at the Sapporo Medical University Hospital, Japan. Regarding the use of human conjunctival fibroblast (HconF) cells, approval was obtained from the institutional review board (IRB registration number 282-8). All experiments were conducted according to the tenets of the Declaration of Helsinki and the national laws for the protection of personal data.

### 2.1. Preparation of 2D Cells and 3D Spheroids Originating from HconF

The preparation of 2D cells and 3D spheroids originating from human conjunctival fibroblasts (HconF, ScienCell Research laboratories, Carlsbad, CA, USA) was processed as essentially described in previous reports [38,39,40,41]. Briefly, the HconF cells were cultured in 150 mm plastic 2D culture dishes using the recommended complete fibroblast culture medium from the cell supplier company (Fibroblast Medium, Cat. #2301) supplemented with fibroblast growth supplement (FGS, Cat. #2352), 2% FBS, and penicillin/streptomycin (P/S, Cat. #0503) at 37 °C and maintained by changing the medium every other day until confluency was reached at 80–90% [38]. Alternatively, those cells were subjected to the 3D spheroid drop culture. That is, after washing them with phosphate-buffered saline (PBS) and following dispersion using 0.25% trypsin/EDTA, the cells were re-suspended in the complete fibroblast culture medium containing 0.25% methylcellulose. A 28 μL aliquot of the suspension containing approximately 20 × 10^5^ HconF cells was added to each well of a 384-hanging drop array plate (# HDP1385, Sigma-Aldrich, St Louis, MO, USA) [42,43], and those 3D spheroid drop cultures were further maintained by exchanging half of the medium each following day until day 6. 

### 2.2. Induction of EMT by the TGF-β Isoforms

For inducing EMT in the 2D and 3D cultured HconF cells, 5 ng/mL of TGF-β-1, -2, or -3 was administered to the fibroblast complete culture medium on days 1 through 6. The dosages of the TGF-β isoforms used in the current study were essentially determined based on previously reported data [38,44]. 

### 2.3. Analysis of Planar Proliferation of the HconF Cell Monolayer by TEER

For the planar proliferation analysis of the 2D cultured HconF cells by a TEER measurement, the HconF cell monolayers were prepared as described in previous studies [38,45]. In brief, the 2D HconF cells cultured in 150 mm 2D culture dishes as described above were re-suspended in the complete fibroblast culture medium after washing with PBS, dispersion by 0.25% trypsin/EDTA, and following centrifugation (5 min at 300× *g*). Approximately 20,000 cells were placed in each well of 12-well plates for TEER (Corning Transwell, Sigma-Aldrich) measurements and cultured until their cell densities reached a confluence of approximately 80%. After administering 5 ng/mL of TGF-β-1, -2, or -3 and maintaining the cultures for 6 days (day 6), the TEER values (Ωcm^2^) of each well were measured as essentially described in previous studies [38,45].

### 2.4. Measurement of Real-Time Mitochondrial and Glycolytic Functions

For estimation of the mitochondrial and glycolytic functions of the 2D cultured HconF cells, their oxygen consumption rate (OCR) and extracellular acidification rate (ECAR) were measured, respectively, using a Seahorse XFe96 Bioanalyzer (Agilent Technologies) as basically described in our previous reports [38,46,47]. Briefly, 20 × 10^3^ 2D cultured HconF cells were seeded onto the wells of a 96-well Seahorse measurement analytical plate administered without (non-treated control, NT) or with solutions of TGF-β-1, -2, or -3 (5 ng/mL). After exchanging the culture medium with Seahorse XF DMEM assay medium (pH 7.4, Agilent Technologies, #103575-100) supplemented with 5.5 mM glucose, 2.0 mM glutamine, and 1.0 mM sodium pyruvate, the basal OCR and ECAR values were determined using a Seahorse XFe96 Bioanalyzer, and thereafter, the samples were further analyzed after supplementation with 2.0 μM oligomycin, 5.0 μM carbonyl cyanide p-trifluoromethoxyphenylhydrazone (FCCP), 1.0 μM rotenone and antimycin A, and 10 mM 2-deoxyglucose (2-DG). The OCR and ECAR values were normalized to the amount of protein per well.

### 2.5. Measurement of the Mechanical Properties, Size, and Solidity of the 3D HconF Spheroids

For evaluation of the mechanical properties, size, and stiffness of the HconF 3D spheroids obtained as above, they were characterized as reported in our previous studies [39,41]. In brief, the mean sizes (μm^2^) of the 3D spheroids were determined as the largest cross-sectional area by using phase contrast microscopy images using a ×4 objective lens and using an inverted microscope (Nikon ECLIPSE TS2; Tokyo, Japan). Alternatively, for the stiffness measurement, a single living 3D spheroid was compressed to its semi-diameter for 20 s using a micro spheroid compressor system (MicroSquisher, CellScale, Waterloo, ON, Canada). The requiring force (μN) and semi-diameter (μm) were simultaneously measured by an equipped micro-pressure sensor and a micro monitoring camera, respectively. To estimate the mechanical stiffness of the 3D spheroid, the index of the requiring force (μN)/semi-diameter (μm) was used.

### 2.6. qPCR Analysis

qPCR analyses were carried out as described in our previous reports [40,48] using specific pre-designed primers and probes (Appendix A). Briefly, total RNA was extracted using an RNeasy mini kit (Qiagen, Valencia, CA, USA), and reverse transcription was performed with the SuperScript IV kit (Invitrogen) in accordance with the manufacturer’s instructions. Respective gene expression was quantified by real-time PCR with the Universal TaqMan Master Mix using a StepOnePlus machine (Applied Biosystems/Thermo Fisher Scientific, Waltham, MA, USA). The quantities of cDNA were normalized to the expression of the housekeeping gene 36B4 (RPLP0) and are shown as the fold change relative to the control.

### 2.7. Statistical Analysis

All statistical analyses were determined using the obtained experimental data expressed by the arithmetic mean ± the standard error of the mean (SEM). To analyze the differences among the matched multiple group comparisons, one-way ANOVA was used followed by a Tukey’s multiple comparison test using Graph Pad Prism 8 software (GraphPad Software, San Diego, CA, USA) as described in our recent reports [40,48].

## 3. Results

### 3.1. Effects of the Three TGF-β Isoforms on the Planar Proliferation of the HconF Monolayers

Initially, to study the effects of the TGF-β isoforms on conjunctival epithelial proliferation, TEER measurements were performed using the 2D cultured HconF cell monolayer. As shown in Figure 1, the TGF-β-2 solution (5 ng/mL) significantly increased the TEER values, consistent with the findings reported in our previous studies [38,46,47]. Similar to TGF-β-2, TGF-β-1 (5 ng/mL) also increased the TEER values, but in contrast, the TEER values were substantially decreased by TGF-β-3 (5 ng/mL). 

### 3.2. Effects of the Three TGF-β Isoforms on the Mitochondrial and Glycolysis Functions of the 2D Cultured HconF Cells

To compare the effects that were induced by the three TGF-β isoforms on the mitochondrial and glycolysis functions of the 2D cultured HconF cells, a real-time cellular metabolic function measurement was performed using a Seahorse XFe96 Bioanalyzer. The results indicated that the addition of the three TGF-β isoforms had no effect on the indices reflecting mitochondrial function and glycolytic function in the 2D HconF cells (Figure 2). These results suggest that the different effects toward the planar proliferation of the HconF cells, as indicated by the TEER measurements for the three TGF-β isoforms, were not related to the metabolic states of the HconF cells.

### 3.3. Effects of the Three TGF-β Isoforms on Subconjunctival Proliferation Using the 3D HconF Spheroids

Next, in order to further examine the inducible influences of the three TGF-β isoforms on subconjunctival proliferation, as the mechanical aspects of the 3D HconF spheroid models, their sizes and stiffness were measured. As demonstrated in Figure 3, those three TGF-β isoforms also affected the spheroids in different manners, that is, TGF-β-1 increased the stiffness, TGF-β-2 did not significantly affect both indices, and TGF-β-3 resulted in down-sized spheroids with increased stiffness.

### 3.4. Effects of the Three TGF-β Isoforms on the Gene Expression of Major ECM Proteins of the 2D and 3D Cultured HconF Cells

To gain additional insight into the diverse effects of the 2D and 3D cultured HconF cells, as observed above, in more detail, a qPCR analysis was performed to determine the mRNA expression levels of some major ECM proteins, including collagen 1 (COL1), COL4, COL6, fibronectin (FN), and α smooth muscle actin (α-SMA), and they were compared under several conditions, that is, in the absence or presence of each TGF-β isoform under 2D and 3D cell culture conditions. In the 2D cultured HconF cells (Figure 4), TGF-β-2 induced significant increases in the mRNA expression of *COL1*, *COL4*, *FN,* and α-*SMA*, and decreases in those of *COL6*, despite the fact of no significant inducible effects by the TGF-β-1 and -3 isoforms. In contrast, in the 3D HconF spheroids (Figure 5), TGF-β-2 substantially upregulated the gene expression of *COL1* and *FN*, but TGF-β-1 induced increases in the expression of *COL6* and α-*SMA*. These collectively observed observations indicate that the different influences toward the expressions of some ECM molecules by the TGF-β isoforms may contribute to the diverse biological aspects in the 2D HconF cells as well as the 3D HconF spheroids. 

## 4. Discussion

To obtain a higher success rate in glaucoma filtering surgery, great interest has developed regarding the use of cytokines or growth factors that are related to the wound healing processes within the eye [49,50]. Among these, TGF-β, a multifunctional growth factor that is a well-known member of a large superfamily of polypeptide molecules [51], is recognized as an important candidate growth factor for regulating the wound healing of several ocular tissues. Among the three isoforms of TGF-β, it has been reported that, of the TGF-β-1, TGF-β-2, and TGF-β-3 that are present within several ocular tissues, TGF-β-2 was identified as the predominant isotype for the cornea [51], conjunctival fibroblasts [52], vitreous and aqueous humor, and tears [53,54]. In fact, it is reported that significantly high concentrations of TGF-β-2 are present in the AH of patients with glaucoma as compared with non-glaucoma subjects [55]. However, recent reports have demonstrated that 1) the AH concentrations of TGF-β-2 are higher in patients with primary open angle glaucoma (POAG) but are lower in patients with pseudo exfoliation glaucoma (PXF), and 2) the AH concentrations of TGF-β-1 and TGF-β-3 are markedly increased in patients with PXF in comparison with other types of glaucoma [29,32,56,57], suggesting that the three TGF-β isoforms may be diversely involved in the pathological mechanisms of several types of glaucoma. Since, in the case of glaucoma filter surgery, the AH permeates into the subconjunctival space called the filtering bleb and its longer survival is exclusively required for better surgical success rates, TGF-β-isoform-induced effects toward conjunctival fibrosis should be a quite important issue. However, as far as we know, these have been insufficiently investigated. In the present study, to compare the effects of conjunctival fibrosis by the three TGF-β isoforms, we used our recently developed in vitro models replicating conjunctival superficial and subepithelial fibrosis using 2D and 3D cell cultures of HconF, respectively [38,46,47]. Regarding the obtained results, these TGF-β isoforms induced different effects toward planar proliferation by TEER and the physical properties of 3D spheroids of HconF cells, suggesting that TGF-β isoforms, especially TGF-β-3, could be differently involved in both epithelial and subepithelial conjunctival fibrosis. 

The application of 3D spheroid cell cultures as suitable in vitro models for various physiological and pathological conditions has recently attracted interest [58]. As compared with conventional 2D cultured cells, 3D spheroid cultured cells have been recognized to be quite different in their biological nature. For instance, the cells within the 3D spheroids could interact with other cells at any location similar to those of in vivo organs as compared with the side-by-side intercellular interactions observed within conventional 2D planar cell cultures. These unique intercellular interactions of the 3D spheroids could also serve more physiologically closer protein networks, including several ECM proteins, cell junction proteins, and others, with those of in vivo organs but distinct from those of the 2D cultured cells. Using these differences in both cell culture systems, we developed different in vitro pathogenic models for conjunctival fibrosis, that is, epithelial and subconjunctival fibrosis by 2D and 3D cultured HconF cells with TGF-β-2, respectively [38,46,47,59]. In the current investigation, these 2D and 3D in vitro culture models using HconF cells in conjugation with various analyses including planar proliferation by TEER, cellular metabolic measurements, and physical property measurements of 3D spheroids, were also quite useful for comparing the three TGF-β isoforms toward conjunctival fibrosis and, therefore, could be applicable to evaluate the efficacy as well as the toxicity of other factors, chemicals, and drugs.

In our recent study, we also compared the effects of the three TGF-β isoforms, TGF-β-1 ~ -3, on human trabecular meshwork (HTM) cells using similar 2D and 3D cell culture models as the present study. Regarding the observed results, we found that (1) all of these TGF-β three isoforms significantly increased the TEER values in the 2D cultured monolayers of the HTM cells, and among these, such effects were most potently exerted by TGF-β-3, (2) TGF-β-3 also caused quite different influences on the mitochondrial and glycolytic metabolic functions in comparison with TGF-β-1 and TGF-β-2, (3) these TGF-β isoforms diversely affected the mechanical aspects of the 3D HTM spheroids, and (4) in the mRNA expression of several related proteins including ECMs and their modulators, the influences of TGF-β-3 were markedly distinct from those of TGF-β-1 and TGF-β-2. Based on these observed results, we suggest that the quite unique and diverse effects of TGF-β-3 toward the HTM in comparison with TGF-β-1 and TGF-β-2 may also result in different effects in the pathogenesis of glaucoma [31]. Similar to the elevated AH concentrations of the TGF-β isoforms, a previous study demonstrated that the intensities of the anti-TGF-β-antibody-labelled conjunctival cells were significantly higher in patients with glaucoma (35.21% ± 14.12%) as compared with control patients (14.96% ± 6.34%) (*p* = 0.001) [60]. In addition, and quite interestingly, such TGF-β-positive conjunctival cells were significantly reduced after a trabeculectomy (23.0% ± 13.8%) (*p* < 0.001), posing the possibility that TGF-β isoforms might be importantly involved in the pathology of glaucoma. Taking these collective findings into account, the results indicate that the three TGF-β isoforms affected planar and subconjunctival proliferation as well as glaucomatous HTM indifferent manners. We therefore conclude that the effects of these TGF-β isoforms toward both conjunctival fibrosis and glaucomatous HTM may lead to a strategy for developing clinical applications designed to prevent pathological conjunctival fibrosis as well as hypotensive therapy in the future. 

However, in contrast to several investigations that focused on the effects of TGF-β-1 and TGF-β-2 on the pathogenesis of glaucoma using HTM cells [32,61,62] and conjunctival fibrosis [37,38,46,47,63,64,65], the influence of TGF-β-3 has not been extensively investigated so far. Therefore, additional studies will be required to elucidate the physiological and pathological significance of these three TGF-β isoforms, especially focusing on TGF-β-3 toward conjunctiva in more detail using RNA sequencing analysis, SiRNA inhibition of the specific target molecules of their downstream signaling, and others as our next project.

## Figures and Tables

**Figure 1 biomedicines-11-02005-f001:**
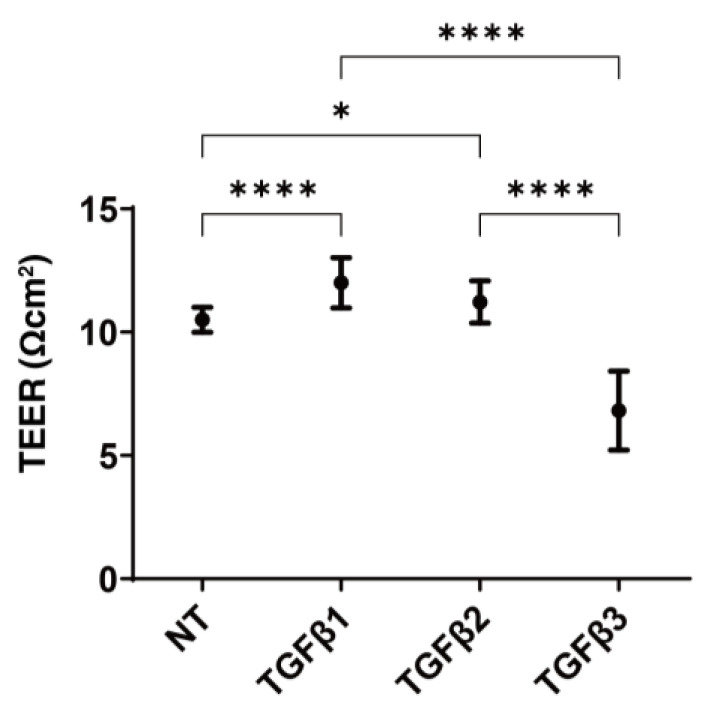
Influence of the TGF-β isoforms, TGF-β-1~-3, on the TEER electrical resistance measurements of the 2D HconF monolayers. The HconF monolayers were prepared by the conventional 2D culture method on day 6 in the absence and presence of TGF-β-1, TGF-β-2, or TGF-β-3 (5 ng/mL). The planar proliferation of those were analyzed by electric resistance (Ωcm^2^) measurements using TEER in triplicate (*n* = 9 each). * *p* < 0.05; **** *p* < 0.001.

**Figure 2 biomedicines-11-02005-f002:**
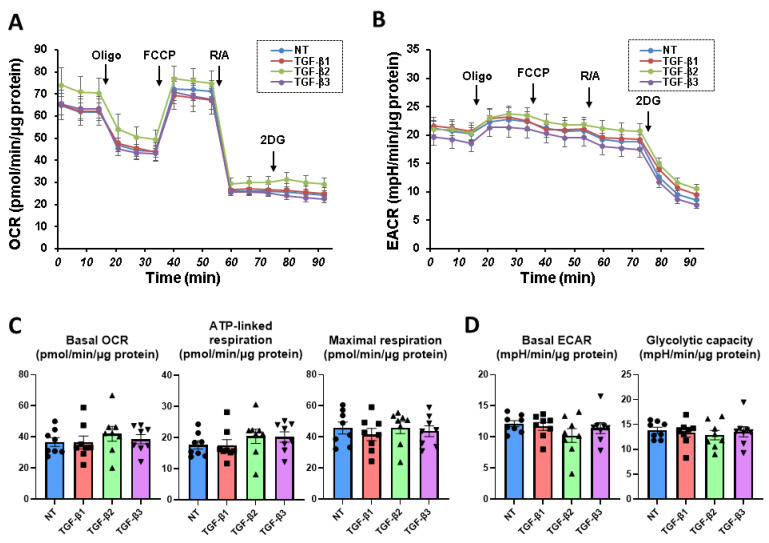
Influences of treatment with the TGF-β isoforms on the mitochondrial and glycolytic functions of the 2D HconF cells. The HconF cells were 2D cultured for 6 days in the absence and presence of TGF-β-1, TGF-β-2, or TGF-β-3 (5 ng/mL, *n* = 8 each) and were subjected to a real-time metabolic function analysis. The OCR (**A**) and the ECAR (**B**) were measured at baseline, and thereafter, they were further measured by subsequent supplementation with oligomycin (Oligo, complex V inhibitor), FCCP (a proton ionophore), and rotenone/antimycin A (R/A, complex I/III inhibitors) and 2-Deoxy-D-glucose (2DG, hexokinase inhibitor). The indices of mitochondrial function (**C**) and glycolytic function (**D**) were calculated as follows: basal respiration: subtracting the OCR with rotenone/antimycin A from the OCR at baseline, ATP-linked respiration: the difference in the OCR after the addition of oligomycin, maximal: subtracting the OCR with rotenone/antimycin A from the OCR with the FCCP, basal ECAR: subtracting the ECAR with 2-DG from the ECAR at baseline, and glycolytic capacity: subtracting the ECAR with 2-DG from the ECAR with oligomycin.

**Figure 3 biomedicines-11-02005-f003:**
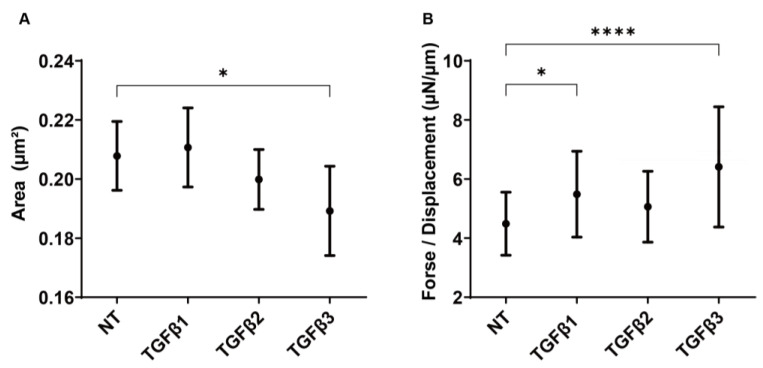
Influence of the TGF-β isoforms, TGF-β-1~-3, on the mechanical properties, sizes, and stiffness of the 3D HconF spheroids. The 3D HconF spheroids were prepared during 6-day cultures in the absence and presence of TGF-β-1, TGF-β-2, or TGF-β-3 (5 ng/mL, *n* = 16 spheroids each). The mean sizes measured are plotted in (**A**). Alternatively, their mechanical solidity was measured by a micro-squeezer, and the requiring force (μN) to compress a single spheroid into its semidiameter (μm) during a period of 20 s is plotted in (**B**). * *p* < 0.05; **** *p* < 0.001.

**Figure 4 biomedicines-11-02005-f004:**
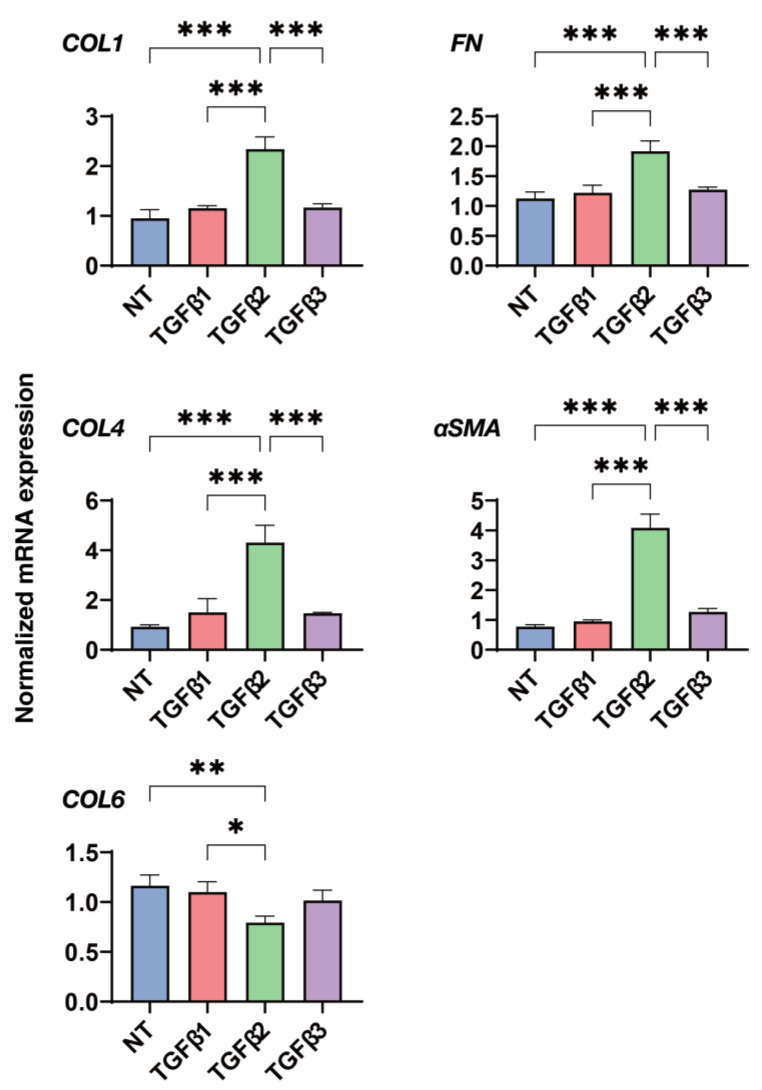
Influences of the TGF-β isoforms, TGF-β-1~-3, on the mRNA expression of ECMs in the 2D cultured HconF cells. The HconF cells were 2D cultured for 6 days in the absence and presence TGF-β-1, TGF-β-2, or TGF-β-3 (of 5 ng/mL, *n* = 5) for qPCR analysis. The mRNA expression of major ECM molecules including *COL1*, *COL4*, *COL6*, *FN,* and *a-SMA* was estimated. * *p* < 0.05; ** *p* < 0.01; *** *p* < 0.005.

**Figure 5 biomedicines-11-02005-f005:**
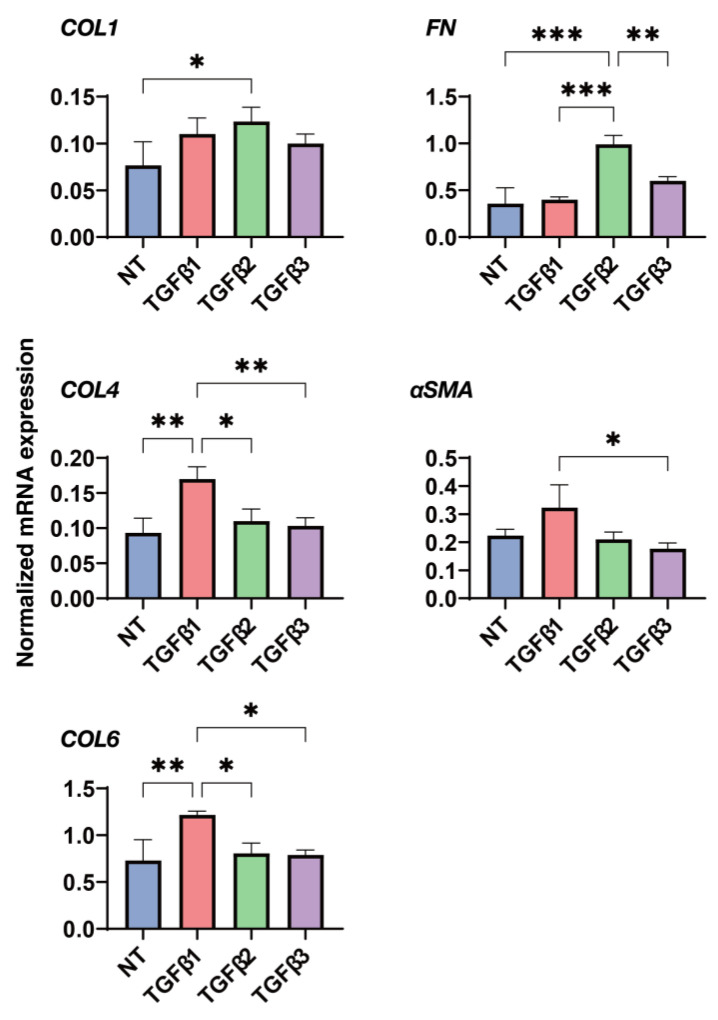
Influence of the TGF-β isoforms, TGF-β-1~-3, on the mRNA expression of ECMs in the 3D cultured HconF cells. The 3D HconF spheroids were prepared during 6-day cultures in the absence and presence of TGF-β-1, TGF-β-2, or TGF-β-3 (5 ng/mL, *n* = 15 spheroids each) for qPCR analysis. The mRNA expression of major ECM molecules including *COL1*, *COL4*, *COL6*, *FN*, and *a-SMA* was estimated. * *p* < 0.05; ** *p* < 0.01; *** *p* < 0.005.

## Data Availability

The data that support the findings of this study are available from the corresponding author upon reasonable request.

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
