# Peer review of "TGF-β Isoforms Affect the Planar and Subepithelial Fibrogenesis of Human Conjunctival Fibroblasts in Different Manners"

_biomedicines, 2023, doi:10.3390/biomedicines11072005_

Round 1

Reviewer 1 Report

Title: TGF-β isoforms affect the planar and subepithelial fibrogene- 2 sis of human conjunctival fibroblasts in different manners

It is interesting study on TGF-β isoforms and mitochondria function in the human conjunctival fibroblasts.

Major concerns

1.     As the tissue is originated from human conjunctiva, although commercial available cell line, it required IRB review and approved waiver since it is experiments.

2.     Line 170, it needed for statistical analysis in detail. ANOVA? Fig 3-5 legend is necessary for statistic method.

3.     Discussion 469-480 the sentences is difficult to understand, it should be re-written for strength of this study results and purposes. If the author intended to compare with the results of previous study, it is make a table for results of previous study.

Minor concerns

Fig 3-C, and Fig 3-4 the bar plot (esp. NT and TGF-β1 and 2 is all most same) is gray, it is better change contrast or color for better understanding. Additionally, Fig 4 and 5 bar plot is all black, it is better to be colored bar.  

English presentation required minor correction.

Author Response

Dear Editor,

Thank you very much for the constructive comments concerning our manuscript, " TGF-β isoforms affect the planar and subepithelial fibrogene- 2 sis of human conjunctival fibroblasts in different manners”. We carefully checked all of the Reviewer comments and prepared a revised version of our paper that takes these comments into account. The changes are listed below. Specific changes within the manuscript are highlighted.

Reviewer 1 comments

Title: TGF-β isoforms affect the planar and subepithelial fibrogene- 2 sis of human conjunctival fibroblasts in different manners

It is interesting study on TGF-β isoforms and mitochondria function in the human conjunctival fibroblasts.

Major concerns

  1. As the tissue is originated from human conjunctiva, although commercial available cell line, it required IRB review and approved waiver since it is experiments.

Answer; Thank you for this comment, As suggested, those requested information was included at very beginning of the Method section; “The current study was conducted at the Sapporo Medical University Hospital, Japan. Regarding the use of human conjunctival fibroblasts (HconF) cells, approval was obtained from the institutional review board (IRB registration number 282-8). All experiments were conducted according to the tenets of the Declaration of Helsinki and the national laws for the protection of personal data.”

  1. Line 170, it needed for statistical analysis in detail. ANOVA? Fig 3-5 legend is necessary for statistic method.

Answer; Thank you for this comment. As suggested, details related to the statistical analyses are now included in the Method section; “All statistical analyses were determined using obtained experimental data expressed by the arithmetic mean ± the standard error of the mean (SEM). To analyze the difference among matched multiple group comparisons, one-way ANOVA was used, followed by a Tukey's multiple comparison test using Graph Pad Prism 8 software (GraphPad Software, San Diego, CA) as described in our recent reports [40,48].”

  1. Discussion 469-480 the sentences is difficult to understand, it should be re-written for strength of this study results and purposes. If the author intended to compare with the results of previous study, it is make a table for results of previous study.

Answer; Thank you for this comment. As suggested, TM tissue is substantially different from conjunctiva and this comparison may create difficulties in terms of understanding our goal here. In addition, another reviewer also asks us to rewrite this paragraph. Therefore, the last paragraph of Discussion was significantly changed; “However, in contrast to several investigations that focused on the effects of the TGF-β-1 and TGF-β-2 on the pathogenesis of glaucoma using HTM cells [32,61,62] and conjunctival fibrosis [37,38,46,47,63-65], the influence of TGF-β-3 has not been extensively investigated so far. Therefore, additional study will be required to elucidate the physiological and pathological significances of these three TGF-b isoforms, especially focusing TGF-b-3 toward conjunctiva in more detail using RNA sequencing analysis, SiRNA inhibition of the specific target molecules of their down-stream signaling and others as our next project.”.

 Minor concerns

  1. Fig 3-C, and Fig 3-4 the bar plot (esp. NT and TGF-β1 and 2 is all most same) is gray, it is better change contrast or color for better understanding. Additionally, Fig 4 and 5 bar plot is all black, it is better to be colored bar.

Answer; Thank you for these comments. As suggested, those figures were revised.

  1. Comments on the Quality of English Language; English presentation required minor correction.

Answer; Thank you for this comment. As suggested, English was carefully edited by a native speaking scientist, Prof. Milton Feather, and a letter of confirmation from him is attached.

Reviewer 2 comments

ID: Biomedicines - 2442727

Type: Article

Title: TGF-β isoforms affect the planar and subepithelial fibrogene-2 sis of human conjunctival fibroblasts in different manners

The work is related to the study of the effect of different TGF isoforms on fibroblasts. It is a well-designed work, with sufficient information. Throughput the paper, the reviewer thinks that some points can be improved to give more strength to the work.

ABSTRACT

  1. The abstract is well written with a clear presentation of the purpose of the research, materials and methods, results and conclusions. Nevertheless, the reviewer thinks that since there are some differences between the different isoforms, a statement presenting the value of that difference concerning treatment, mechanism of pathology or other feature could be present.

Answer; Thank you for this excellent comment. As suggested, at first, this information is included; Three, highly homologous isoforms of TGF-b, TGF-b-1 ~3 are involved in the regulation of various pathophysiological conditions such as wound healing processes in different manners, despite the fact that they bind to the same receptors during their activation.”.

INTRODUCTION

  1. The introduction is well written with a clear presentation of the problematic studied in this work, as well as the approach presented. Nevertheless, the reviewer would suggest a more detailed description on the different therapeutic possibilities of the different isoforms and some examples if there were some.
  2. The value of this work is the characterization of the effect of the different isoforms in conjunctiva, so the reviewer suggest that the authors state since the introduction the differences of these isoforms and their potential. Additionally, the authors say that the EMT of fibroblasts into myofibroblasts is important. Do you have any work on this differentiation? Why did you not follow that differentiation in this paper?

Answers for #2 and #3; Thank you for these excellent comments. As suggested, the requested information is now included in the last part of the 3rd paragraph; “For example, among the TGF-b isoforms, previous studies have revealed that 1) TGF-b-1 induces collagen production in the conjunctival fibroblast that is mediated by NADPH oxidase4 (Nox4) related signaling [33], 2) the amniotic membrane matrix uniquely suppresses the production of  TGF-b-2 and -3 [34], 3) TGF-β signaling inhibits goblet cell differentiation via the SAM-pointed domain that contains the ETS transcription factor (SPDEF) in the conjunctival epithelium [35], and 4) TGF-β signaling play a significant profibrotic roles in the pterygium via stimulating EMT [36,37]. Thus, these collective observations suggest that further investigations concerning the functional diversity among these TGF-b isoforms may facilitate, not only a better understanding of pathophysiological conjunctival fibrosis, but also would provide some clues for developing new therapeutic strategies for the treatment of pathogenic conjunctival fibrosis.”.

  1. Line 41, “extracellular matrix” – extracellular matrix (ECM)

Answer; Thank you for this comment. As pointed out, those were changed to “extracellular matrix (ECM)”

  1. Line 42, “believed” – change the word

Answer; Thank you for this comment. As pointed out, that was changed to “shown”.

  1. Line 64, “epithelial mesenchymal transition (EMT) – a “ is missing

Answer; Thank you for this comment. As pointed out, this was changed to “epithelial mesenchymal transition (EMT)”.

  1. Line 73, “Transforming growth factor β (TGF-β)” – already described (line 62)

Answer; Thank you for this comment. As pointed out, this repetition was changed.

  1. Line 89, typo

Answer; Thank you for this comment. As pointed out, this typo error was corrected.

MATERIALS AND METHODS

  1. Overall, the information here is sufficient; however, the reviewer suggest some modifications on the information for each section, and the addition of the number of assays performed in each section. The reviewer thinks that a study on the effect of the different isoforms on the EMT of fibroblasts into myofibroblast would be a plus, since the authors say that this transition is important.

Answer; Thank you for this comment. As suggested, the TGF-b isoform treatment information within the cell culture method was moved a new section related to EMT; “Induction of EMT by TGF-b isoforms

For inducing EMT in 2D and 3D cultured HconF cells, a 5 ng/mL TGF-β-1, -2 or -3 was administered to their fibroblast complete culture medium at Days 1 through 6. The dosages of the TGF-β isoforms used in the current study were essentially conducted based on previously reported data [38,44]”.

  1. Line 101, the reviewer suggest that occurs keep in this section just the information related to cell culture and maintenance of 2D and 3D models. The information on TEER, real time mitochondria, and drug treatment must go to the respective section.

Answer; Thank you for this comment. As suggested, the information concerning TEER, real time mitochondria, and drug treatment was deleted within the cell culture preparation.

  1. Line 106, “form” – from

Answer; Thank you for this comment. As pointed out, that was changed to “from”.

  1. Line 107, with the addition of any supplement?

Answer; Thank you for this comment. As pointed out, supplement was missing, and therefore this information is now included, “complete fibroblast culture medium from the cell supplier company (Fibroblast Medium, Cat. #2301) supplemented with fibroblast growth supplement (FGS, Cat. #2352), 2% FBS and penicillin/streptomycin (P/S, Cat. #0503)”.

  1. Line 110, “TEER” – already described

Answer; Thank you for this comment. As pointed out, this repetition was removed.

  1. Line 129, initial cell density is required

Answer; Thank you for this comment. As pointed out, more detail including initial cell density was included; “Approximately 20,000 cells were placed in each well of 12-well plates for TEER (Corning Transwell, Sigma-Aldrich) measurements and cultured until their cell densities reached at a confluence of approximately 80 %.”.

  1. Line 132, the reviewer suggests that all the information on the TEER study must be added here.

Answer; Thank you for this comment. As pointed out, information related to TEER within the cell culture method was removed and all information is now described here.

  1. Line 139, “basically” – delete this word

Answer; Thank you for this comment. As pointed out, “basically” was deleted.

  1. Line 152, “hardness” – or stiffness?

Answer; Thank you for this comment. As pointed out, this term was unified as “stiffness”.

  1. Line 155, in which microscope, with which objective?

Answer; Thank you for this comment. As pointed out, those detail was included “using a ×4 objective lens using an inverted microscope (Nikon ECLIPSE TS2; Tokyo, Japan)”.

  1. Line 163, more information on the conditions are missing

Answer; Thank you for this comment. As pointed out, more detailed information is now included; “qPCR analyses were carried out as described in our previously reports [40,48] using specific pre-designed primers and probes (supplemental Table 1). Briefly, total RNA was extracted using an RNeasy mini kit (Qiagen, Valencia, CA), and reverse transcription was performed with the SuperScript IV kit (Invitrogen) in accordance with the manufacturer’s instruction. Respective gene expression was quantified by real-time PCR with the Universal TaqMan Master Mix using a StepOnePlus machine (Applied Biosystems/Thermo Fisher Scientific). The quantities of cDNA were normalized to the expression of housekeeping gene 36B4 (RPLP0) and are shown as fold change relative to the control.”.

  1. Line 164, “essentially” – delete this word

Answer; Thank you for this comment. As pointed out, “essentially” was deleted.

  1. Line 170, “essentially” – delete this word

Answer; Thank you for this comment. As pointed out, “essentially” was deleted.

 RESULTS

  1. This section is a little confused. The reviewer understands the idea of not adding subsections to make the text more fluid, but the reviewer would suggest dividing it into TTER, metabolic, spheroids, qPCR.

Answer; Thank you for this excellent comment. As suggested, the results were divided into TEER, metabolic, spheroids and qPCR with appropriate subheadings.

  1. Line 174, this information is not a result.

Answer; Thank you for this comment. As suggested, these were removed.

  1. Line 176, this information is not a result.

Answer; Thank you for this comment. As suggested, these were removed.

  1. Line 242, “nardness” – typo

Answer; Thank you for these comments. As suggested, those were properly change to “hardness”.

DISCUSSION

  1. This section can be improved. The authors start with spheroids, and then go to the isoforms and then the clinical outcomes of the isoforms. The reviewer would suggest to start with the problem, detailed the solution, presenting the tools used to study that and finally the clinical outcomes. Additionally, the first and second paragraphs are too long. In contrast, the clinical implications of the authors’ results are too short. The work is good, so the reviewer thinks that it must be highlighted in a better way in this section.

Answer; Thank you for this excellent comment. As suggested, we started with the problem, detailed the solution, presenting the tools used to study that and finally the clinical outcomes in the discussion as much as possible.

  1. Line 395, “outstanding” – remove this word.

Answer; Thank you for this comment. As suggested, this has now been removed.

  1. Comments on the Quality of English Language; Some typos. Some paragraphs are too long

Answer; Thank you for this comment. As suggested, English was carefully edited by a native speaking scientist, Prof. Milton Feather, and a letter of confirmation from him is attached.

Reviewer 2 Report

Report

ID: Biomedicines - 2442727

Type: Article

Title: TGF-β isoforms affect the planar and subepithelial fibrogene-2 sis of human conjunctival fibroblasts in different manners

The work is related to the study of the effect of different TGF isoforms on fibroblasts. It is a well-designed work, with sufficient information. Throughput the paper, the reviewer thinks that some points can be improved to give more strength to the work.

ABSTRACT

The abstract is well written with a clear presentation of the purpose of the research, materials and methods, results and conclusions. Nevertheless, the reviewer thinks that since there are some differences between the different isoforms, a statement presenting the value of that difference concerning treatment, mechanism of pathology or other feature could be present.

INTRODUCTION

The introduction is well written with a clear presentation of the problematic studied in this work, as well as the approach presented. Nevertheless, the reviewer would suggest a more detailed description on the different therapeutic possibilities of the different isoforms and some examples if there were some.

The value of this work is the characterization of the effect of the different isoforms in conjunctiva, so the reviewer suggest that the authors state since the introduction the differences of these isoforms and their potential. Additionally, the authors say that the EMT of fibroblasts into myofibroblasts is important. Do you have any work on this differentiation? Why did you not follow that differentiation in this paper?

Line 41, “extracellular matrix” – extracellular matrix (ECM)

Line 42, “believed” – change the word

Line 64, “epithelial mesenchymal transition (EMT) – a “ is missing

Line 73, “Transforming growth factor β (TGF-β)” – already described (line 62)

Line 89, typo

MATERIALS AND METHODS

Overall, the information here is sufficient; however, the reviewer suggest some modifications on the information for each section, and the addition of the number of assays performed in each section. The reviewer thinks that a study on the effect of the different isoforms on the EMT of fibroblasts into myofibroblast would be a plus, since the authors say that this transition is important.

Line 101, the reviewer suggest that occurs keep in this section just the information related to cell culture and maintenance of 2D and 3D models. The information on TEER, real time mitochondria, and drug treatment must go to the respective section. 

Line 106, “form” – from

Line 107, with the addition of any supplement?

Line 110, “TEER” – already described

Line 129, initial cell density is required

Line 132, the reviewer suggests that all the information on the TEER study must be added here.

Line 139, “basically” – delete this word

Line 152, “hardness” – or stiffness?

Line 155, in which microscope, with which objective?

Line 163, more information on the conditions are missing

Line 164, “essentially” – delete this word

Line 170, “essentially” – delete this word

RESULTS

This section is a little confused. The reviewer understands the idea of not adding subsections to make the text more fluid, but the reviewer would suggest dividing it into TTER, metabolic, spheroids, qPCR.

Line 174, this information is not a result.

Line 176, this information is not a result.

Line 242, “nardness” – typo

DISCUSSION

This section can be improved.

The authors start with spheroids, and then go to the isoforms and then the clinical outcomes of the isoforms. The reviewer would suggest to start with the problem, detailed the solution, presenting the tools used to study that and finally the clinical outcomes. Additionally, the first and second paragraphs are too long. In contrast, the clinical implications of the authors’ results are too short. The work is good, so the reviewer thinks that it must be highlighted in a better way in this section.

Line 395, “outstanding” – remove this word.

Some typos.

Some paragraphs are too long.

Author Response

(The authors gave the same response as above.)
